# Dietary Intervention during Weaning and Development of Food Allergy: What Is the State of the Art?

**DOI:** 10.3390/ijms25052769

**Published:** 2024-02-27

**Authors:** Alessandro Gravina, Francesca Olivero, Giulia Brindisi, Antonia Fortunata Comerci, Chiara Ranucci, Cinzia Fiorentini, Eleonora Sculco, Ethel Figliozzi, Laura Tudini, Viviana Matys, Daniela De Canditiis, Maria Grazia Piccioni, Anna Maria Zicari, Caterina Anania

**Affiliations:** 1Department of Maternal Infantile and Urological Science, Sapienza University of Rome, 00161 Rome, Italy; alessandro.gravina@uniroma1.it (A.G.); giulia.brindisi@uniroma1.it (G.B.); antoniafortunata.comerci@uniroma1.it (A.F.C.); chiara.ranucci@uniroma1.it (C.R.); cinzia.fiorentini@uniroma1.it (C.F.); ethel.figliozzi@uniroma1.it (E.F.); laura.tudini@uniroma1.it (L.T.); viviana.matys@uniroma1.it (V.M.); mariagrazia.piccioni@uniroma1.it (M.G.P.); annamaria.zicari@uniroma1.it (A.M.Z.); 2Independent Researcher, 00161 Rome, Italy; francescaolivero31@gmail.com; 3Department of Translation and Precision Medicine, Sapienza University of Rome, 00161 Rome, Italy; eleonora.sculco@uniroma1.it; 4Institute of Applied Calculus-CNR Rome, 00185 Rome, Italy; daniela.decanditiis@gmail.com

**Keywords:** food allergy, weaning, FA in weaning, early introduction, egg allergy

## Abstract

Food allergy (FA) affects approximately 6–8% of children worldwide causing a significant impact on the quality of life of children and their families. In past years, the possible role of weaning in the development of FA has been studied. According to recent studies, this is still controversial and influenced by several factors, such as the type of food, the age at food introduction and family history. In this narrative review, we aimed to collect the most recent evidence about weaning and its role in FA development, organizing the gathered data based on both the type of study and the food. As shown in most of the studies included in this review, early food introduction did not show a potential protective role against FA development, and we conclude that further evidence is needed from future clinical trials.

## 1. Introduction

Food allergy (FA) represents a significant public health problem that has emerged over the last 10–15 years, and has increased in prevalence over recent decades [1]. The clinical approach to FA has always been controversial. In the past, the prevention of FA was based on food evasion. In high-risk infants (i.e., those with atopy in first-degree family members), certain foods were recommended to be avoided in pregnancy and during breastfeeding to prevent the onset of allergic diseases [2]. Additionally, the major scientific societies used to recommend a late introduction of allergenic foods in weaning for children who were at risk of developing allergies [3]. These recommendations came from the assumption that the “immaturity” of the mucosal barrier during infancy would allow an easier sensitization to food antigens [4]. Despite these recommendations, several studies through the years have shown that a late exposure to allergenic foods not only did not reduce the risk of FA [5,6,7] but also could lead to severe repercussions for patients’ and families’ quality of life, increasing costs for society [8].

Therefore, current guidelines issued by several scientific societies, such as the World Health Organization (WHO), the American Academy of Pediatrics (AAP) [9], the European Academy of Allergy and Clinical Immunology (EAACI) [10], the European Society for Pediatric Gastroenterology Hepatology and Nutrition (ESPGHAN) [11], and the European Food Safety Authority (EFSA) no longer recommend this approach, based on the fact that a delayed introduction of solid food (after 6 months of age) has been demonstrated to be non-protective against FA development both in high- and low-risk infants. On the contrary, clinical practice guidelines now recommend an early introduction of allergenic foods, changing infant feeding practices in some regions [1]. It remains unclear whether earlier introduction of allergenic foods will reduce overall FA prevalence in the population. Several studies have reported reduced FA following earlier introduction of multiple allergenic foods, particularly in peanut and egg allergy [2,3,4]. The aim of this narrative review is to collect and summarize all the studies that concern the effects of an early introduction of potential allergens during weaning in infants, drawing conclusions about the possible role of weaning in preventing FA development in both high- and low-risk infants.

### 1.1. Food Allergy Immuno-Pathogenesis

FA can be defined as an adverse immunological reaction in susceptible subjects to a specific food antigen that is normally non-dangerous to the healthy population [12]. Chickens’ eggs, cow’s milk, peanuts, tree nuts, fish, crustacean shellfish, wheat and soy are implicated most [13]. Food allergies can be caused by two different immunological mechanisms: the IgE-mediated and the non-IgE-mediated ones. The former are characterized by the presence of IgE antibodies specific to a certain allergen, which are formed by B cells after their stimulation consequent to antigen presentation by the antigen-presenting cells (APCs) to T cells. Upon re-exposure to the same allergen, histamine and other mediators are released and this results in immediate hypersensitivity allergy symptoms [14,15]. The non-IgE-mediated forms of FA are a minority of the total, are less often characterized and are usually due to acute or chronic inflammation in the gastrointestinal (GI) tract [16,17]. The clinical manifestations of food allergies include skin, GI, respiratory and systemic manifestations with varying severity [18,19,20]. The diagnostic approach begins with a careful medical history collection to identify the culprit food, and a physical examination [21]. Oral food challenge (OFC) currently represents the diagnostic gold standard [22]. Treatment consists of the elimination of the offending allergen. Patients at risk of anaphylaxis should be trained to recognize early symptoms and should be instructed in the proper use of self-injectable epinephrine [23]. FA prevention strategies have changed in recent years and current evidence shows that there is a time window for food introduction, including allergy-causing food (both in infants at high risk for atopy and in the general population). Early introduction of allergenic foods in infancy has emerged as one of the more promising strategies to decrease FA development.

The purpose of this narrative review is to collect all the studies that describe the effects of early introduction of potential allergens during weaning in infants at high risk of developing FA and in the general population.

### 1.2. Digestion and Mucosal Barrier

Through the GI tract, the ingested food undergoes a series of degradation processes that allow the absorption of nutrients [24]. The first step in proteins’ degradation occurs in the mouth [25]. Once the denaturized proteins reach the small intestine they are subjected to both endopeptidase and exopeptidase activity [26] in the epithelial brush border [27].

The state in which dietary proteins are absorbed into the intestinal lumen is fundamental to their allergenicity. Many factors influence the digestion of food proteins and among these are the processing of the foods and the matrix in which they are cooked [24,28,29].

The digestive process can be influenced by different factors, such as enzymatic activity, gastric pH, health status, diet and GI microbiota [17,18].

The intestinal barrier represents one of the first and most important defense mechanisms and it plays a role in the pathophysiology of food allergies. One of its main purposes is to maintain intestinal homeostasis, limiting allergens’ access and permitting the passage of nutrients to the subepithelial space. This protective function can be ensured thanks to the two-layer composition of the intestinal barrier: an external physical layer and an internal immunological layer [30].

The former is the epithelial monolayer, composed of enterocytes, goblet cells, enteroendocrine cells, Paneth cells and stem cells located at the crypt’s base [31].

Enterocytes, goblet cells and Paneth cells play a key role in the pathogenesis of FA. Epithelial cells secrete eotaxin-1, a chemo-attractant for eosinophil cells. Goblet cells with their mucus layer provide a physical and chemical protective barrier against enteric bacteria, by acting as a molecular filter, limiting bacterial penetration into the mucous membranes. They also regulate the balance between food tolerance and allergy, playing a fundamental role in the antigen transfer of immune cells from the lumen to the lamina. Paneth cells, located at the base of the intestinal crypt, secrete numerous peptides (AMPs) such as lysozyme, which protect the integrity of the epithelial barrier, maintaining a homeostatic balance with the microbiota [32].

In healthy subjects the paracellular transition of elements is well regulated by the presence of junctional complexes between enterocytes [33]. Above the epithelial layer there is the mucus layer, which is composed of high O-glycosylated glycoproteins secreted by the prior mentioned goblet cells [34,35,36]. This layer plays both a protective role against GI microbes and permits the passage of nutrients and essential molecules [23,24].

The internal immunological layer is composed primarily of immune cells that reside within the intestinal epithelium. The innate immune system represents the first line of the immunological layer; the prime mechanism is the activation of the pattern-recognition receptors (PRRs), such as the toll-like receptors (TLRs). These receptors are responsible for the activation of the innate immune cells and the release of antimicrobial elements and inflammatory cytokines [37]. When the innate immune system is not sufficient, the adaptive system is activated. The main actors of the adaptive immune system are the lymphocytes, which is also thanks to the secretion of sIgA, while the other mechanism is mediated by the oral tolerance [38].

The above-mentioned barrier works differently in allergic people when compared to healthy subjects [39]. It is unclear whether FA is responsible for the increased permeability of the mucosal barrier or whether individuals with high intestinal permeability are at greater risk of developing food allergies. Several studies have been conducted on this topic, and have concluded that both theories could be valid [30]. The other mechanism that is possibly responsible for the onset of food allergies is the different uptake that characterizes the atopic subjects [40,41]. An allergic reaction is triggered when intact proteins or their fragments pass through the epithelial layer. Studies conducted on rats and mice show that in the allergic models, the passage of bigger fragments or entire proteins through the epithelial layer happens exclusively via the transcellular pathway, facilitating the formation of the IgE/CD3 complex [29,30].

According to the dual exposure hypothesis, oral tolerance is achieved through food ingestion, while the development of allergy is the consequence of exposure to the allergen through the skin [42,43].

Atopic subjects present an enhanced transport that is composed of two phases. Phase I occurs thanks to a higher concentration of IgE CD3 receptors (FcεRII) on the enterocytes of allergic subjects and happens when dietary allergens bind to the IgE/CD3, guaranteeing their transport to the lamina propria. Phase II starts once dietary allergens are released by the IgE/CD3 complex into the lamina propria and bind to the IgE already attached to the tissue resident mast cells, inducing a degranulation response. The mediators released by these mast cells (histamine, prostaglandins and proteases) affects the junctional complexes (Figure 1) [39].

Epithelial damage or inflammation (e.g., due to exposure to toxins or trauma) in the intestine allows increased antigen entry and promotes secretion of the epithelial-derived cytokine interleukin-25 (IL-25), IL-33 and thymic stromal lymphopoietin (TSLP). These mediators “set” the immune system towards a T helper 2 (TH2) cell response. In particular, TSLP can promote the differentiation of dendritic cells (DCs) into a TH2 cell-promoting phenotype. Secretion of IL-25 can also promote the expansion of populations of innate lymphoid cell type 2 (ILC2), which together with TH2 cells secrete cytokines. They promote the TH2 cell-mediated immune response, which includes the accumulation of eosinophils in tissues and IgE class switching by B cells [31,44].

### 1.3. Microbiota and Food Allergies

Our organism is colonized by an enormous number of micro-organisms. Most of them live in our gut, constituting the so-called gut microbiota. The human gut microbiota represents a composite ecosystem that contributes to crucial functions for the host, such as promoting metabolic benefits, ensuring immune homeostasis and immune responses, and protecting against pathogen colonization [31,32,33,34]. Its alteration increases the risk of FA development.

Gut microbiota composition can be influenced by different factors, such as dietary habits, the use of formulas instead of breastfeeding, antibiotic therapies, and others. Its composition undergoes substantial modifications during two life phases in which the diet faces important changes: from birth to weaning and from weaning to adulthood.

Interestingly, the factors acting on the early colonization process strongly influence the post-weaning colonization pattern. Early diversification, as observed under formula-feeding not containing prebiotics, promotes earlier acquisition of an adult-type microbiota. Further diversification of diet gradually increases diversity and abundance of Bacteroidetes and Firmicutes towards adult levels, and generally low abundant levels of [45]. The influence of early colonization patterns on the composition of the adult microbiome is not yet fully understood. However, these patterns have been shown to influence gut maturation, immune development and host metabolism [46,47]. Differences in composition driven by environmental factors in infancy may affect susceptibility to metabolic (e.g., obesity), immunologic (e.g., IBD and allergy) and even behavioral (e.g., autism) disorders into adulthood [48].

The relationship between the host and the microbiota is symbiotic and can maintain a stable environment in which microorganisms are fed with nutrients supplied in the intestinal tract. Meanwhile, the host acquires products from microbial fermentation conversion of dietary indigestible components, i.e., fibers, into short-chain fatty acids (SCFA).

SCFAs play an essential role in reaching immune-tolerance to food antigens. SCFAs include butyrate, propionate, acetate and pentanoate. All these are critical to the immune homeostasis of the gut. Butyrate promotes an anti-inflammatory pathway via GRP109A, which stimulates DCs to produce IL10 by inducing the differentiation of CD4+ T cells peripherally, and regulatory T (Treg) cells [49].

SCFA contributes to an estimated 10% of our energy requirement, vitamin K and B12 production, and acts as a defense against potential pathogens. An alteration of gut microbiota can influence the development of several diseases, such as functional GI diseases, infections, IBD, liver diseases, GI malignancies, obesity and metabolic syndrome, diabetes mellitus, autism, allergic diseases and others (Figure 2) [45,48].

Studies conducted on gnotobiotic (germ-free) mice demonstrated that a *Clostridia*-containing microbiota can play a protective role in terms of FA development, and its presence can be altered by the use of antibiotics. [30,50,51,52].

In germ-free mouse models, the lack of microbial colonization results in a failure of the gut-associated lymphoid tissues (GALT) development, promoting a Th2 skewed immune response. Microbial colonization is a major event in the development of Th1 response and Treg cells, which are responsible for the maintenance of immunologic balance and promotion of tolerance. Moreover, gut microbiota plays a main role in the development and maintenance of barrier function and its alteration may promote allergic sensitization [30,53].

Gut microbiota is responsible for the activation of TLR in intestinal epithelial cells (IECs). TLR4-deficient mice are more liable to develop FA due to a Th2 tilted immune response. Finally, it has been demonstrated that mice with a gain-of-function mutation in the IL-4 receptor α chain present a specific microbiota signature that results in an increased liability to oral allergic sensitization and anaphylaxis [34,54,55].

It has recently been shown that children affected by FA present higher abundances of *Ruminococcus gnavus* and *Faecalibacterium prausnitzii* and a reduction of other species belonging to the [45] Bifidobacterium and Bacteroides families [56]. This composition of gut microbiota increases the suppressive activity of Tregs cells and induces the production of IL-10 from Foxp3 + T cells [57].

There is evidence that alteration in gut bacterial population (known as dysbiosis) can be a hypothesis of FA predisposition [53,58].

**Figure 2 ijms-25-02769-f002:**
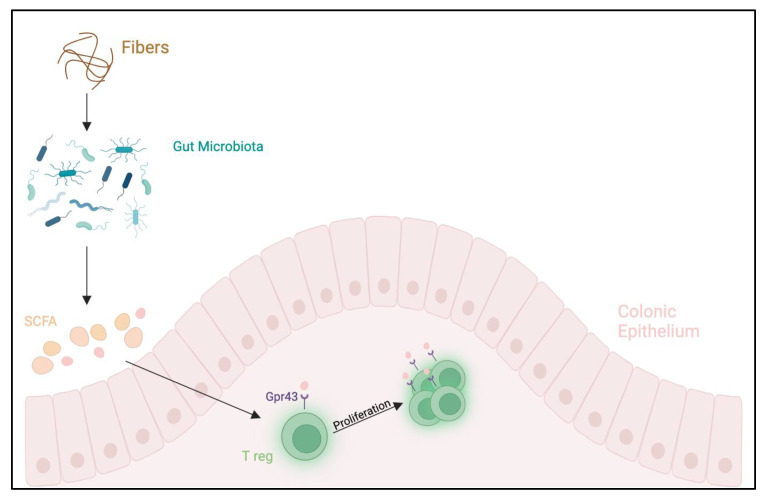
Various microrganisms, known as gut microbiota, colonize human intestine lumen. These microorganisms are responsible for the production of SCFA (acetate, butyrate, propionate), which bind specific G protein-coupled receptors (GPRs) on the IEC’s surfaces; for example, GPR43. Recently, it has been demonstrated that these receptors play a main role in inflammation control, GI functions, allergy development and other manifestations [59]. Created with biorender.com/ (accessed on 12 January 2024).

## 2. Method

To assess the role of weaning in the prevention of FA, we analyzed several studies, evaluating the possible consequences of early food introduction (i.e., eggs, cow’s milk, peanuts and others). Most of these studies were performed in high-risk populations. Patients are considered at high risk of developing FA if they have a history of eczema or if they have first-degree relative with history of eczema, asthma, allergic rhinitis or FA. To improve the value of the study, we also included studies on general populations; in this way, it is easier to assess the general role of weaning in the development of FA.

In this narrative review, a comprehensive search was conducted using MEDLINE via PubMed, Uptodate and Scopus dashboard. After the selection, all studies were carefully analyzed, to reach conclusions regarding the actual knowledge in this field. The following keywords were used for the research: “food allergy”, “weaning”, “FA in weaning”, “early introduction”, “egg allergy”. Studies from 2008 to 2023 were selected, excluding reviews and meta-analyses.

## 3. Studies

Several studies have focused on the relationship between early food introduction and FA prevention in high-risk and in general populations (Table 1).

### 3.1. Studies Performed in High-Risk Populations

#### 3.1.1. Egg Proteins

Regarding hen’s egg, five intervention studies have been conducted; among these, four were performed in high-risk populations and showed no clear reduction of allergic manifestations [55,56,57,58,59].

However, one study conducted in Japan showed a significant reduction of hen’s egg allergy without adverse events: the Prevention of Egg Allergy with Tiny Amount Intake Trial (PETIT). It is a randomized, double-blind, placebo-controlled trial (RDBPCT), conducted by Natsume, et al. in 2017, as a two-step egg introduction study whose goal is to understand whether an early introduction of eggs, combined with optimal eczema treatment, can be effective in prevention of egg allergy in high-risk infants with eczema, aged 4 to 5 months. A total of 147 infants were randomly assigned to early introduction of egg or placebo. The primary outcome was the proportion of participants with a positive OFC to hen’s egg at 1 year of age, assessed blindly by standardized methods, in all randomly allocated participants who received the intervention. The results showed that 5 (8%) of 60 participants had an egg allergy in the egg group, compared with 23 (38%) of 61 in the placebo group (RR (95% CI):0.22 (0.090–0.543); *p* = 0.0001). The authors concluded that the introduction of heated egg in a stepwise manner along with aggressive eczema treatment is a safe and efficacious way to prevent hen’s egg allergy in high-risk infants [66].

A RDBPCT was performed by Palmer, et al. in 2013 to determine whether infants with moderate-to-severe eczema would benefit from early regular oral egg exposure, in terms of IgE-mediated egg allergy reduction. A total of 86 infants, who had never tried egg, were allocated either to the group that received 1 teaspoon of pasteurized raw whole egg powder (*n* = 49) or rice powder (*n* = 37) daily from 4 to 8 months of age. The primary outcome was to investigate the onset of IgE-mediated egg allergy at 12 months. A high proportion, 31% (15/49), of infants assigned to the study group had an allergic reaction to the egg powder and did not continue powder ingestion. At 4 months of age, before any known egg ingestion had occurred, egg-specific IgE were already present in 36% (24/67) of infants. At 12 months, a diagnosis of IgE-mediated egg allergy was given to a lower (but not significant) proportion of infants in the study group (33%) compared to the control group (51%; relative risk, 0.65; 95% CI, 0.38–1.11; *p* = 0.11). The authors concluded that an early, regular oral exposure to egg in infants with moderate–severe eczema may induce immune tolerance [61].

Again, Palmer et al. in 2017 conducted a RCT in which infants aged 4 to 6 months at hereditary risk of developing hen’s egg allergy were randomly allocated to receive daily pasteurized raw whole egg powder (study group, n = 407) or a color-matched rice powder (control group, n = 413) up to age 10 months. In both groups, all infants followed a strict egg-free diet; cooked egg was introduced to the both study and control groups at age 10 months. The primary outcome was IgE-mediated egg allergy defined by a positive pasteurized raw egg challenge and egg sensitization at age 12 months. This study concluded that there is no significant difference between the two studied groups: egg group 7.0% vs. control 10.3%; adjusted relative risk, 0.75; 95% CI, 0.48–1.17; *p* = 0.20 [67].

In 2017, in an Australian study by Tan et al., the authors conducted an RCT on high-risk infants aged between 4 and 6 months to assess whether dietary introduction of egg in this age range would reduce sensitization to egg (Beating Egg Allergy Trial, BEAT). Infants with a skin prick test (SPT) response to egg white (EW) of less than 2 mm were randomized at age 4 months to receive whole-egg powder or placebo (rice powder) until 8 months of age. An EW SPT response of 3 mm or greater at age 12 months represented the primary outcome of the study. Of the 319 infants enrolled, 165 were randomized to the egg group, and 154 to the placebo group. SPT of 3 mm or greater to EW at 12 months was present in 20% of infants in the placebo group and in 11% of infants in the egg group. The study concluded that early oral introduction of whole-egg powder in high-risk infants reduces sensitization to EW and induced egg-specific IgG4 levels. RCTs addressing the preventive effect of the early introduction of egg have provided mixed and conflicting results, which are potentially due to differences in the study population outcomes and study design, including the form of egg used (i.e., raw vs. cooked) [65].

#### 3.1.2. Cow’s Milk, Peanut, Hard-Boiled Hen’s Egg, Sesame, Whitefish (Cod) and Wheat

A recent RCT conducted in Japan in 2022 evaluated the possible role of simultaneous administration of very small amounts of multiple foods in preventing multiple food allergies. The study group enrolled 163 infants aged 3 to 4 months from 14 Japanese primary care pediatric clinics; all the infants were diagnosed with atopic dermatitis. The infants assigned to the study group (*n* = 83) received mixed allergenic food powder (MP) composed of egg, cow’s milk, wheat, soybean, buckwheat and peanuts, while placebo powder (PP). was given to infants assigned to the control group (*n* = 80). The amount of powder was increased in a stepwise manner on weeks 2 and 4 and continued until week 12. The occurrence of FA episodes after powder intervention was assessed at 18 months of age. A significant difference emerged between the MP and the PP group, in terms of FA episodes incidence, by 18 months (7/83 vs. 19/80, respectively; risk ratio 0.301 [95% CI 0.116–0.784]; *p* = 0.0066). The occurrence of egg allergy was lower in the MP group. Furthermore, all the other analyzed foods caused a significantly reduced number of FA episodes [68].

#### 3.1.3. Rusk-like Biscuit Powder

A protocol for an RDBPCT, single-center clinical trial is ongoing in Berlin. In the trial, 150 infants aged between 4 and 8 months, all with atopic eczema, will be randomized in a 2:1 manner into an active group that will receive rusk-like biscuit powder with hen’s egg (HE), cow’s milk (CM), peanuts (PN) and hazelnuts (HN) for 6–8 months. In the placebo group, a sugar-free rusk-like biscuit powder without the above-mentioned allergens, will be provided, analogously to the intervention group, daily. The proportion of allergens in the powder will be increased by three times every 6 weeks. The infants sensitized to HE, CM, PN or HN at the end of the interventional period will undergo an OFC. The primary endpoint is to determine the occurrence of IgE-mediated FA to at least one of the four foods after 6–8 months of intervention (i.e., at 1 year of age). The secondary endpoints include the occurrence of multiple food allergies, severity of eczema, wheezing and sensitization levels to food allergens [69].

#### 3.1.4. Peanut

Du Toit started the Learning Early About Peanut Allergy (LEAP) study in 2015 in England, which included 640 babies aged between 4 and 11 months, who had severe eczema or an allergy to egg, or both. Patients were grouped into two different study cohorts based on the results of a skin-prick test for peanut; participants in each study cohort were then randomly assigned to a group in which dietary peanut would be consumed or a group in which its consumption would be avoided. Among the 530 infants in the intention-to-treat population, who initially had negative results on the SPT, the prevalence of peanut allergy at 60 months of age was 13.7% in the avoidance group and 1.9% in the consumption group (*p* < 0.001). Among 98 participants in the intention-to-treat population, who initially had positive test results, the prevalence of peanut allergy was 35.3% in the avoidance group and 10.6% in the consumption group (*p* = 0.004). This is the first RCT study demonstrating that early introduction of peanut modulates the allergic response in children at high risk of developing this type of FA [62].

### 3.2. Studies Performed in Low-Risk Populations

#### 3.2.1. Egg Proteins

In 2017, Bellach, et al. conducted an RDBPCT to assess the possible role of early hen’s egg introduction in preventing sensitization (Hen’s Egg Allergy Prevention, HEAP). A total of 383 infants not sensitized to hen’s egg at 4 to 6 months of age were randomized to the placebo or the intervention group. First, 524 infants were enrolled in the study but, in the end, 298 patients were evaluated for the primary outcome, showing that 5.6% (6/124) in the study group and 2.6% (4/152) in the placebo group were sensitized to hen’s egg at age 12 months (*p* 0.35); in addition, 2.1% were confirmed to have hen’s egg allergy versus 0.6 in the placebo group. In this study there was no evidence that early consumption of hen’s egg prevents hen’s egg allergy [64].

#### 3.2.2. Cow’s Milk, Peanut, Hard-Boiled Hen’s Egg, Sesame, Whitefish (Cod) and Wheat

In 2016 Perkin, et al. conduced the Enquiring About Tolerance (EAT) study which is a population-based randomized controlled trial that enrolled exclusively breastfed infants from England and Wales regardless of atopic status or family history of allergy to assess whether the early introduction of six common foods (including peanuts) could prevent FA in a general population. The early introduction group (EIG) continued breastfeeding with sequential introduction of 6 allergenic foods between 4 and 6 months of age: cow’s milk, peanut, hard-boiled hen’s egg, sesame, whitefish (cod) and wheat; the standard introduction group (SIG) continued exclusive breastfeeding for around 6 months. A total of 1303 infants were enrolled. The control about tolerance were based on the execution of SPTs, blood exams, eczema (SCORAD score), microbiota, growth and a food diary, at 1 and 3 years of age. According to the intention-to-treat analysis, 7.1% infants of the SIG (42 of 595) and 5.6% of those in the EIG (32 of 567) developed allergy to at least one of the six intervention foods [63].

### 3.3. Studies Performed in Both High-Risk and Low-Risk Populations

#### Cow’s Milk Proteins

In 2008, Sneijder conducted a prospective cohort study evaluating the association between the introduction of cow’s milk products or other solid food products and atopic manifestations in the second year of life. A total of 2558 infants were enrolled and data on the introduction of cow’s milk products and other food products, and on the outcomes (eczema, atopic dermatitis, recurrent wheeze, any sensitization, sensitization against cow’s milk, hen’s egg, peanut and at least one inhalant allergen), were collected through questionnaires at 34 weeks of gestation and at 3, 7, 12 and 24 months postpartum. They showed that a delayed introduction of both cow’s milk products and other food products was associated with a higher risk for eczema (*p* = 0.01 and 0.02 for trend, respectively), while a delay in other food products’ introduction was associated with a higher risk for AD according to UK-WP criteria (*p* = 0.00 trend) [60].

## 4. Discussion

FA represents a significant burden, affecting financial, social and health aspects. Therefore, their prevention would have a significant impact. The possible role of an early introduction of food in subjects at high and low risk of developing FA has been investigated in the literature. Studies conducted on high-risk populations showed reduced allergic manifestations when an allergen was introduced early during weaning. However, the same results were not obtained in the general population. Several trials have analyzed the relationship between the early introduction of eggs into the diet and the risk of egg allergy. In Bellach’s study, the introduction, 3 times a week, of 2.5 g of pasteurized white egg before the age of 6 months, in children not at risk of developing allergy, did not decrease egg allergy onset [68]; in contrast to this aspect, the remaining four studies, conducted in high-risk populations, conversely showed a reduction in egg allergy when the food was introduced early into the diet [68]. Similarly, Du Toit and his group demonstrated that an early introduction of peanuts during weaning can reduce the incidence of allergic manifestations related to the culprit food in a high-risk population [65]. Overall, the evidence concerning the FA prevention towards a particular allergen by its early introduction, shows statistical significance for eggs and peanuts in the high-risk population only. The lack of uniformity in the methodology of the various published studies, as well as in the different enrolled populations, may be responsible for bias that affects the final results.

In the EAT [63] and KOALA [60] studies, no statistical difference was found in the general population after the early introduction of various allergenic foods (such as eggs, cow’s milk, peanuts and whitefish). The same results were found in the Kalb study [69], although conducted in a pediatric population with eczema and at high risk of developing FA.

The quantity, frequency and number of allergenic foods could influence the development of food allergies and the results of these studies.

Further studies could clarify not only the optimal time for introducing food but also which types of children could benefit from the early introduction of food allergens.

## 5. Conclusions

Evidence shows that there is a time window for food introduction, including allergy-causing food for inducing oral tolerance. Consequently, the recommendations of major scientific societies have changed since 2008, and many FA prevention guidelines now recommend the early introduction of allergenic food, such as peanuts and eggs, as part of complementary feeding in infancy.

However, the available studies suggest that the early administration of some food positively influences high-risk populations. Meanwhile, low-risk populations do not benefit from the same treatment. Further studies are needed, especially on the general population, to be able to provide indications in this regard.

Other elements, such as the form in which the allergen is introduced and the continuity of its intake, play an essential role in preventing FA. However, at the moment, the data obtained make it challenging to develop universal guidelines. Further studies are needed to clarify the timing of food introduction and understand whether weaning can play a central role in FA prevention in both high- and low-risk populations.

## Figures and Tables

**Figure 1 ijms-25-02769-f001:**
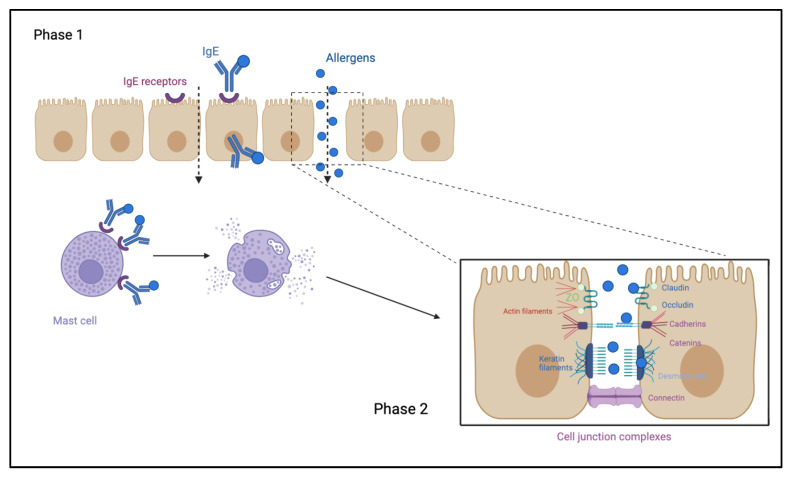
Increased protein uptake in allergic individuals. The IgE receptor CD23 is upregulated on the enterocytes surface of atopic subjects and the formation of the complex IgE/CD23 after allergen exposure allows for passage through the epithelial layer (phase 1). Once the allergens get in contact with the subepithelial mast cells, the degranulation response is induced. The mediators released by the mast cells modify the cells junctional complex, increasing the paracellular space and protein uptake (phase 2). Created with biorender.com/ (accessed on 12 January 2024).

**Table 1 ijms-25-02769-t001:** Main clinical studies evaluating the possible role of weaning in preventing food allergy development in infants.

Author, Year, Country, Trial	Study Design	Sample Size	Population	Inclusion Criteria	Allergen	Outcome	Main Results
Snijders et al., 2008 Netherlands KOALA [60]	Prospective birth cohort study	2558 infants	General population	Pregnant women with diverse lifestyles	Cow’s milk products and other solid products.	Questionnaires at 7, 12, 24 months; Specific IgE >0.3 UI/mL against eggs, cow’s milk at age 2	Delayed introduction of CMP and other food products associated with higher risk for eczema (*p* = 0.01 and 0.02 for trend, respectively); no association between delayed introduction of CMP and AD; delayed introduction of other food associated with higher risk for AD (*p* = 0.00 trend) and increased risk of atopy development at the age of 2 years
Palmer et al., 2013 Australia STAR [61]	RDBPCT	86 infants: 49 SG 37 CG	High risk	4 months of age singleton term infants with moderate-to-severe eczema no prior egg or solid food ingestion	Hen’s egg (0.9 g/day)	OFC and SPT at 12 months	At 12 months 33% SG, 51% CG were diagnosed IgE-mediated egg allergy (relative risk, 0.65; 95% CI, 0.38–1.11; *p* = 0.11)
Du Toit et al., 2015 England LEAP [62]	RCT	640 infants: 319 SG 321 CG	Infants 4 to 11 months of age with severe eczema, egg allergy, or both	High risk	Peanut (6 g/week)	Open OFC or DBPCFC at 12, 30 and 60 months	In the intention-to-treat population: 13.7% in the CG and 1.9% in the SG who had negative SPT developed peanut allergy (*p* < 0.001)
Perkin et al., 2016 England EAT [63]	RCT	1303 infants: 652 SG 651 CG	Exclusively breastfed infants for ≥4–5 months, regardless of atopic status or family history of allergy	General population	Cow’s milk, peanut, hard-boiled Hen’s egg, sesame, whitefish (cod) and wheat at 3 and 6 months of age (4 g/week)	OFC at 1 and 3 years of age after allergenic food introduction	Among infants with sensitization to 1 or more foods at enrollment, EIG infants developed significantly less FA than SIG infants Intention to treat: SIG, 7.1%; EIG, 5.6% *p* = 0.32 Per protocol: SIG, 7.3% EIG, 2.4% *p* = 0.01
Bellach et al., 2017 Germany HEAP [64]	RDBPCT	383 infants: 184 SG 199 CG	GA ≥ 34 weeks and birth weight ≥ 2.5 kg Specific IgE to egg <0.35 kU/L	General population 4–6 months	Hens’ egg 2.5 g 3 times/week from 4–6 to 12 months	OFC and specific IgE ≥0.35 KU/L at 12 months after hen’s egg introduction	Sensitized to hen’s egg at age 12 months: 5.6% (6/124) in SG 2.6% (4/152) in CG (*p* = 0.35); allergy to hen’s egg 2.1% in SG 0.6% in CG (relative risk,3.30;95%CI, 0.35–31.32; *p* = 0.35); no prevention in hen’s egg sensitization nor egg allergy
Tan et al., 2017 Australia BEAT [65]	RDBCT	319 infants: SG 165 CG 154	Infants with at least 1 first-degree relative with allergic disease and SPT < 2 mm	High risk	Hens’ egg 350 mg from 4–8 months	EW SPT response of 3 mm or greater OFC to whole egg at age 12 months.	Sensitization to EW at 12 months: 20% in CG 11% in SG; allergy to EW at 12 months: 10.5% in CG 6.2% in SG (odds ratio, 0.46;95% CI, 0.22–0.95; *p* = 0.03)
Natsume et al., 2017 Japan PETIT [66]	RDBPCT	147 infants	4–5 months of age with eczema	High risk	Eggs 50 mg/die (3–9 months) 250 mg/die (9–12 months)	Open OFC at 12 months of age	Five (8%) of 60 participants had an egg allergy in the SG compared to 23 (38%) of 61 in the CG (risk ratio 0.221;95% CI, 0.090–0.543; *p* = 0.0001)
Palmer et al., 2017 Australia STEP [67]	RCT	820 infants: SG 165 CG 154	Singleton infants with atopic mothers, recruited before age 6.5 months No prior egg ingestion and allergic disease	High risk	Hens’ egg pasteurized raw whole egg powder (SG = 407) or a rice powder (CG = 413) from 6 to 10 months; introduction of egg at 10 months	OFC to egg at 12 months and SPT positive	At 12 months: IgE-mediated food allergy: SG 7.0% vs. CG 10.3% (RR(95%CI) 0.75 (0.48–1.17) *p* = 0.20)
Nishimura et al., 2022 Japan SEED [68]	RCT	163 children: 83 SG 80 CG	3–4 months old with atopic dermatitis	High risk	Egg, milk, wheat, soybean, buckwheat, and peanuts. Amount of powder increased at 2, 4 and 12 week.	The occurrence of FA at 18 months old	Incidence of FA episodes by 18 months: SG 7/83 vs. CG 19/80; (risk ratio 0.301 [95% CI 0.116–0.784]; *p* = 0.0066). Egg allergies were reduced in the SG group
Kalb et al., 2022 German TEFFA [69].	RCT	150 infants with atopic eczema at 4–8 months randomized in a 2:1 manner into an SG and CG	4–8-month-old infants with eczema	High risk	Rusk-like biscuit powder with HE, CM, PN, HN 2 mg for 6–8 months	After 6 months of intervention they will check sensitization against hen’s egg, cow’s milk, hazelnut and peanut	At 12 months egg allergy: SG 2.1% CG 0.6% (3.30;95% CI, 0.31–3132 *p* = 0.35)

KOALA (in Dutch): Child, Parent and Health: Lifestyle and Genetic Constitution; CMP: cow milk proteins; AD: atopic dermatitis; STAR = Solid Timing for Allergy Research; RDBPCT = randomized double-blind placebo-controlled trial; SG: study group; CG: control group; LEAP = Learning Early About Peanut; RCT = randomized controlled trial; EAT = Enquiring About Tolerance; OFC = Oral Food Challenge EIG: Early Introduction Group; SIG: Sequential Introduction Group; HEAP = Hens’ Egg Allergy Prevention; GA: gestational age; BEAT = Beating Egg Allergy Trial; EW: egg white; SPT = skin prick test; PETIT = Prevention of Egg Allergy with Tiny Amount Intake Trial; STEP = Starting Time of Egg Protein; SEED = Start Eating Early Diet; FA = Food Allergy; TEFFA = Early Feeding to Prevent Food allergy in Infant with Eczema; HE: hen’s egg; PN: peanuts: HN: hazelnuts.

## Data Availability

Not applicable.

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
