# Peer review of "Dietary Intervention during Weaning and Development of Food Allergy: What Is the State of the Art?"

_ijms, 2024, doi:10.3390/ijms25052769_

Round 1
Reviewer 1 Report
Comments and Suggestions for Authors
1、2.1. Digestion: It should be supplemented whether the digestive process (such as enzymatic activity, gastric pH, subject health status, diet and GI microbiota) is related to allergic reactions.
2、2.2. Mucosal barrier: composition and role: Indeed, the intestinal barrier is related to the degree of allergic reaction. But how does this part of the author's description relate to early food introduction and weaning?
3、 3. Microbiota and food allergies:Similarly, the authors should discuss the association between early food introduction or weaning and microbial structure, and describe the relationship with the development of allergies.
4、Figure 2: The authors have drawn the picture here to illustrate the importance of SCFA for the progression of immunity and allergies. However, the description of how SCFA affects allergic reactions is too general.
5、Parts II and III should be linked to the theme.
Minor editing of English language required
Author Response
see the uploaded file

Reviewer 2 Report
Comments and Suggestions for Authors
Following are some suggestions that could be used to make this review more informative:
· The section on digestion and mucosal barrier provides a clear understanding of the physiological processes involved in food digestion and the role of the intestinal barrier in preventing food allergen entry. However, It might be helpful to include a brief mention of recent studies or findings related to the modulation of the gut microbiota and its influence on food allergy development, as this is an area of growing interest and relevance.
· Furthermore, the article could benefit from the incorporation of transcriptional-based studies elucidating the roles of key cellular components including enterocytes, goblet cells, enteroendocrine cells, Paneth cells, and mast cells, and their intricate interactions in the pathogenesis of food allergy. By integrating findings from such studies, a deeper understanding of the molecular mechanisms underlying allergic responses during the weaning process can be attained, thereby enriching the quality and comprehensiveness of the information presented.
· Moreover, discussing the involvement of specific molecular pathways, such as the Th1/Th2 balance, regulatory T cell function, and epithelial barrier integrity, could provide valuable insights into the dynamic interplay between genetic predisposition, environmental factors, and immunological responses in the development of food allergy during early life stages. This comprehensive approach would not only enhance the scientific rigor of the review but also facilitate the identification of potential targets for therapeutic interventions aimed at mitigating the risk of food allergy in susceptible individuals.

Comments on the Quality of English Language· The frequent use of "Thanks to" throughout the article could be refined to enhance professionalism and clarity.
Author Response
se the uploaded file

Reviewer 3 Report
Comments and Suggestions for Authors
In this manuscript, the authors reviewed recent research works concerning weaning and its role in food allergy development with a purpose to elucidate the relationship between weanning and food allergy development. The manuscript was generally well organized and easy to understand. However, its major drawback was lack of novelty and extremely few data was delivered by the authors themselves. It is recommeded that during writing the manuscript, the authors should obtain more solid data to make the manuscript more scietifically.
This review intends to understand the effect of early introduction of potential allergens during weaning in infants on the risk of developing food allergy. Though the review is well written, unfortunately, it is not publishable in its current form, and reconsideration after major revision is recommended.
Firstly, from the structure of this review, the submitted review should be a systematic review. In this case, authors provided too much background information (including sections 1, 2, 3, and 4), while the description of methods and discussion (i.e., sections 5 and 6) was too concise. Therefore, it is recommended that sections 1, 2, 3, and 4 can be combined and then shortened as the part of introduction, since in these sections, only some basic information about food allergy is introduced. In addition, more details of the literature search strategy, paper selection, data extraction, statistical analysis should be added in the section of methods. Moreover, in the section of results (i.e, the section 7), authors only simply listed and described the results of different studies. It is lack of explanation of findings in each research (especially the underlying mechanisms of action) and general discussion on similar or different conclusions in different studies. More importantly, as a systematic review, the extracted data from different papers should be analyzed by a meta-analysis, which has not been included in this review.
Secondly, the title should also be corrected. The expression of ‘weaning and development of food allergy’ is vague and nonspecific. Since the aim of this review is to explore whether the earlier introduction of allergenic foods can reduce overall FA prevalence in population, the title can be changed into ‘Dietary intervention during weaning and development of food allergy: what is the state of the art?’.
Lately, this review has poor language quality, such as containing many long and complex sentences (e.g., lines 127-132, 217-224) and grammar mistakes (e.g., it’s (its) in line 124, fails (fail) in line 146, ≥ ? months, 4-5 months of age of age, and 06% in table 1), such that it cannot be understood by readers.
Author Response
see the uploaded file

Reviewer 4 Report
Comments and Suggestions for Authors
The aim of this review manuscript is to summarize all the studies regarding the effects of an early introduction of potential allergens during weaning in infants to demonstration of food allergy symptoms. The authors provide at first a description of the mechanisms involved in food allergy and the implication of the gut microbiota to food allergy. The discussion of the studies is organized on the basis of the allergic foods.
The manuscript is well organized and very well written. It can be therefore read from non experts that search information about the subject of food allergy. The results of the studies discussed are also summarized in a comprehensive way in a Table, that helps the readers to compare the different studies. For these reasons I recommend publication of the manuscript in its current form.
Author Response
see the uploaded file

Reviewer 5 Report
Comments and Suggestions for Authors
The authors of the publication made a narrative review of the 10 most relevant articles that appeared in the global medical literature in 2008-2022. These publications present the results of early introduction of potential allergens (eggs, cow's milk, peanuts, and others) during weaning, in infants at high risk of developing food allergies, as well as among infants and young children representing the general population
The results of the analyzed studies concerned infants and young children from Europe: Netherland, England, Germany (5 publications) and the pediatric population from Australia and Japan (5 publications).
In a methodological procedure (RCT, RDBPCT, prospective cohort study), the researchers assessed the effects of early introduction of allergens with proven allergen potential, giving infants and young children single food allergens (hen's egg proteins, cow's milk proteins, peanuts) or simultaneously very small amounts of multiple foods (cow's milk, peanut, hard boiled hen's egg, sesame, whitefish code and wheat).
Varied results of this research are discussed in detail in the Discussion section. They should be considered as the current state of the art regarding the issues specified in the title of the publication.
Important information that helps explain the process of sensitization of the human body by food allergens is included in Chapter 2 Digestion and mucosal barriers and the 2 figures therein (Fig.1, Fig.2)
The authors of the publication conclude that at the moment current data obtained make it difficult to develop universal guidelines, concerning weaning and development of food allergy in children.
In the reviewer's opinion, the difficulties in obtaining consistent, repeatable results to develop an appropriate universal position are caused by: the ethnic diversity of the studied populations, their regional eating habits, the economic status of the families of the studied children, the type of allergic process (IgE-dependent, non-IgE-dependent, mixed) and diversified research methodology.
The reviewed publication contains new scientific and educational values.
Author Response
see the uploaded file
Round 2
Reviewer 3 Report
Comments and Suggestions for Authors
This manuscript has been well revised based on the reviewers' comments and should be accepted for publication in IJMS as it is.